# Use of Banana Waste as a Source for Bioelectricity Generation

Segundo Rojas-Flores [1],*, Magaly De La Cruz-Noriega [1], Renny Nazario-Naveda [2], Santiago M. Benites [2], Daniel Delfín-Narciso [3], Luis Angelats-Silva [4] and Emzon Murga-Torres [4]

1 Instituto de Investigación en Ciencias y Tecnología de la Universidad Cesar Vallejo, Trujillo 13001, Peru; mdelacruzn@ucv.edu.pe
2 Vicerrectorado de Investigación, Universidad Autónoma del Perú, Lima 15842, Peru; scored731@gmail.com (R.N.-N.); santiago.benites@autonoma.pe (S.M.B.)
3 Grupo de Investigación en Ciencias Aplicadas y Nuevas Tecnologías, Universidad Privada del Norte, Trujillo 13007, Peru; daniel.delfin@upn.edu.pe
4 Laboratorio de Investigación Multidisciplinario, Universidad Privada Antenor Orrego (UPAO), Trujillo 13008, Peru; langelatss@upao.edu.pe (L.A.-S.); ee_mt_66@hotmail.com (E.M.-T.)
* Correspondence: srojasfl@ucvvirtual.edu.pe

**Abstract:** The large amounts of organic waste thrown into the garbage without any productivity, and the increase in the demand for electrical energy worldwide, has led to the search for new eco-friendly ways of generating electricity. Because of this, microbial fuel cells have begun to be used as a technology to generate bioelectricity. The main objective of this research was to generate bioelectricity through banana waste using a low-cost laboratory-scale method, achieving the generation of maximum currents and voltages of 3.71667 ± 0.05304 mA and 1.01 ± 0.017 V, with an optimal pH of 4.023 ± 0.064 and a maximum electrical conductivity of the substrate of 182.333 ± 3.51 μS/cm. The FTIR spectra of the initial and final substrate show a decrease in the peaks belonging to phenolic compounds, alkanes, and alkenes, mainly. The maximum power density was 5736.112 ± 12.62 mW/cm$^2$ at a current density of 6.501 A/cm$^2$ with a peak voltage of 1006.95 mV. The molecular analysis of the biofilm formed on the anode electrode identified the species *Pseudomonas aeruginosa* (100%), and *Paenalcaligenes suwonensis* (99.09%), *Klebsiella oxytoca* (99.39%) and *Raoultella terrigena* (99.8%), as the main electricity generators for this type of substrate. This research gives a second use to the fruit with benefits for farmers and companies dedicated to exporting and importing because they can reduce their expenses by using their own waste.

**Keywords:** banana waste; bioelectricity; microbial fuel cells; organic waste

## 1. Introduction

The demographic growth of society has led to an exponential increase in food consumption in different areas (livestock, vegetable, vegetables, etc.), which has caused problems for governments and companies dedicated to the sale and distribution of food-related products [1,2]. On the one hand, population growth indeed brings great economic benefits to traders, but it also generates greater environmental pollution due to the increase in waste generated in the process of selling and consuming food [3,4]. Government agencies in many countries, mainly in low-income countries (Peru, India, Brazil, Haiti, etc.), do not have an adequate system for the collection and disposal of the waste generated, and often the organic waste from fruits or vegetables is dumped in the adjoining areas of large food centers [5,6]. According to Di Fonzo et al. (2021), the average organic waste generated globally in 2018 was 2 billion metric tons per year and it is estimated that by 2050 this value will increase by 3400 million tons [7]. Similarly, it has been reported that the average annual waste generated by a person in the years 2017, 2018, and 2019 amounted to 33, 47.3, and 55.6 kg, respectively [8]. This has led research centers to become involved in this area to provide a novel solution to all types of waste for the good of society or companies; the waste is currently used to generate fertilizers [8], bioremediation [9], ethanol [10], bioenergy [11],

and others. Bioenergy is gaining importance because it mostly uses organic wastes as fuel sources for its production. For example, Katiyar et al. (2019) studied the particular case of bagasse (sugarcane waste), mentioning that with 5 752,800 metric tons it can generate 9475 GWh per year of bioenergy and, according to the regulations of the government of Pakistan, bagasse can be converted into 2000 MW of electric power [12].

In this sense, microbial fuel cells (MFCs) are a technology that uses organic wastes directly as fuel sources. The generation of electricity is based on the process of oxidation and reduction in the anodic and cathodic chambers that make up the cells through the electrodes and the electrical circuit of such cells [13,14]. Currently, many works use MFCs for electricity generation; for example, Florian et al. (2019) used banana and orange peel wastes, managing to generate voltage peaks of 0.67 V in cells made with activated carbon and zinc electrodes for a volume of 100 mL of orange waste for 10 days [15]. Similarly, Varma and Bebber (2019) used fruit sediment wastes (mango, orange, and banana) as substrates in their cells fabricated with felt electrodes, managing to generate voltage peaks of 370, 130, and 120 mV for cells with orange, mango, and banana substrates, respectively. Orange, tangerine, and lime wastes have also been used as substrates to generate electricity, achieving peaks between 0.95 and 1 V for each individual cell, but using zinc and copper metal electrodes [16].

On the other hand, the production of bananas worldwide has increased exponentially; due to this, countries such as Ecuador, Colombia, Peru, and Brazil, mainly in South America, have increased their exports of this fruit. In 2016, they produced approximately 126 million tons worldwide, representing 15% of fruit consumption. In Brazil alone, 101,992,743 tons were produced, achieving a turnover of US$ 28,209,561 thousand dollars [17,18]. In general, bananas are a rich source of fiber, minerals (phosphorus, magnesium, zinc, potassium), vitamins (C, B6, provitamin A), and phenolic compounds, becoming a functional food [19]. However, its consumption has led to an increase in waste, causing environmental problems in many countries due to the lack of organization for the collection of this waste [20].

Because of this, the main objective of this research is to generate electrical energy using banana waste as fuel (substrate) in laboratory-scale microbial fuel cells, in which the generated values of voltage, current, pH, conductivity, and degrees Brix will be monitored; the power density, current density, internal resistance of the cell, and the initial and final compounds will also be calculated by Fourier-transform infrared spectroscopy (FTIR). Likewise, the microorganisms adhered to the anodic biofilm that generates electricity will be identified through the molecular technique.

## 2. Materials and Methods

### 2.1. Collection of Banana Waste

Five kilograms of decomposing bananas were collected from La Hermelinda market, Trujillo, Peru, which were taken in airtight bags to the laboratory. The waste was washed 5 times with distilled water, in order to eliminate any type of dirt (dust, mud, or insects) acquired from the environment where they were found, and left to dry in an oven at $25 \pm 2$ °C for 12 h. The banana pulp went through an extractor (Labtron, LDO-B10-USA) in order to obtain juice from the waste. It was possible to obtain 1.5 L of juice, which was placed in a beaker and stored until its use in the microbial fuel cells.

### 2.2. Construction of Microbial Fuel Cells

Three (3) low-cost microbial fuel cells were constructed using an acrylic tube (Poly-methyl methacrylate) of 20 × 5 cm in length and diameter, respectively, as an anode chamber. Zinc (Zn) electrodes were used as anode and Copper (Cu) as cathode, each one with an area of 80 cm$^2$. The anode was placed 5 cm from the end of the tube and the cathode was placed at the tube cap to be in contact with the environment (O$_2$) and the substrate at the same time. The electrodes were connected through an external circuit with a resistor of 100 Ω and in the absence of a proton-exchange membrane, see Figure 1.

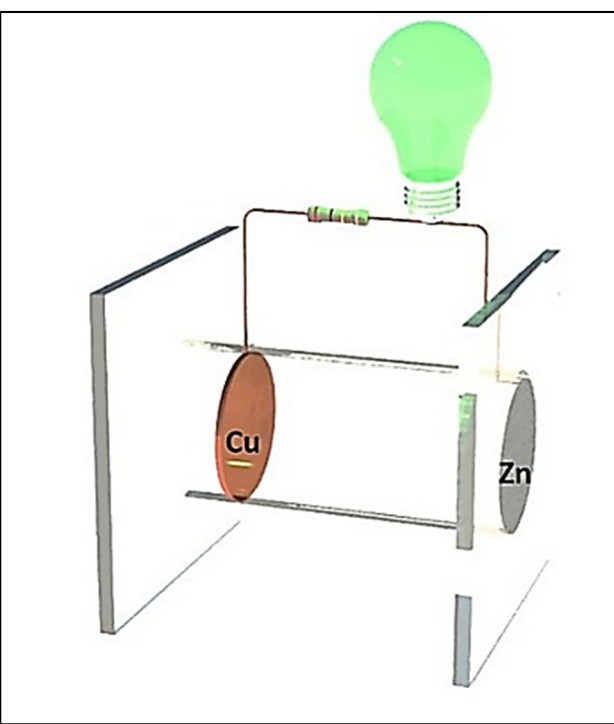

**Figure 1.** Microbial fuel cell prototype.

### 2.3. Characterization of Microbial Fuel Cell

The values of the physical-chemical parameters were measured for 35 days, using a multimeter (Prasek Premium PR-85). It was possible to obtain voltage and current values with an external resistor of 1000 Ω. Conversely, for current density (DC) and power density (PD) values, the procedure performed by Rojas-Flores et al. (2021) was used, with external resistors of 0.3 (±0.1), 0.6 (±0.18), 1 (±0.3), 1.5 (±0.31), 3 (±0.6), 10 (±1.3), 20 (±6.5), 50 (±8.7), 60 (±8.2), 100 (±9.3), 120 (±9.8), 220 (±13), 240 (±15.6), 330 (±20.3), 390 (±24.5), 460 (±23.1), 531 (±26.8), 700 (±40.5), and 1000 (±50.6) Ω [6]. Changes in conductivity (conductivity meter CD-4301), pH (pH meter 110Series Oakton), and degrees Brix (RHB-32 Brix refractometer) were also measured. Transmittance values were measured by FTIR (Thermo Scientific IS50) and the resistance values of the MFCs were measured by using an energy sensor (Vernier-± 30 V and ±1000 mA).

### 2.4. Isolation of Electrogenic Microorganisms from the Anodic Chamber

The electrogenic bacteria were isolated by using the streaking technique from a swabbing in the following growth media: Trypticase Soy agar, Mac Conkey agar, and nutrient agar incubated at 36 °C to isolate Gram-negative bacteria; and Sabouraud agar for fungi and yeasts incubated at 30 °C. The procedure was performed in duplicate [21,22].

### 2.5. Molecular Identification of Bacteria and Fungi

Molecular identification was performed by the Analysis and Research Center of the laboratory "Biodes Laboratorios". From pure cultures, DNA extraction using the CTAB technique, PCR amplification, and 16S rRNA sequencing process was performed by the MACROGEN Laboratory and then analyzed by the bioinformatics software MEGA X (Molecular Evolutionary Genetics Analysis). Finally, the sequence obtained was compared with the sequences of reference bacterial species contained in the genomic base banks, using the sequence alignment tool, BLAST (Basic Local Alignment Search Tool), to obtain the percentage of identity in the identification of bacteria [23].

### 3. Results

Figure 2a shows the values of the voltages generated during the 35 days of monitoring the microbial fuel cells. It is observed that the voltage values increase from the first day ($0.923 \pm 0.005$ V) to the sixth day ($1.01 \pm 0.017$ V) and then slowly decay until the last day ($0.2875 \pm 0.0252$ V). According to Hassan et al. (2019), the increase in voltage values is mainly due to the formation of the bacterial biofilm and the transfer of electrons between the electrodes [24]. For this substrate, after the seventh day, the decay phase begins, which can originate so quickly due to the copper electrode which, although an excellent conductor of electrons, could harm certain microorganisms due to its toxicity [25]. Figure 2b shows the values of the electric currents generated during monitoring, which increased abruptly from the first day ($2.9612 \pm 0.035$ mA) to the fourth day ($3.71667 \pm 0.05304$ mA) of monitoring, and then declined continuously until the last day ($1.45 \pm 0.0932$ mA). Previous studies have shown that the decrease in electrical current values is due to the consumption of the substrate (banana waste) by bacteria that do not generate electricity [26], as well as the depletion of glucose and mannitol by the substrate [27].

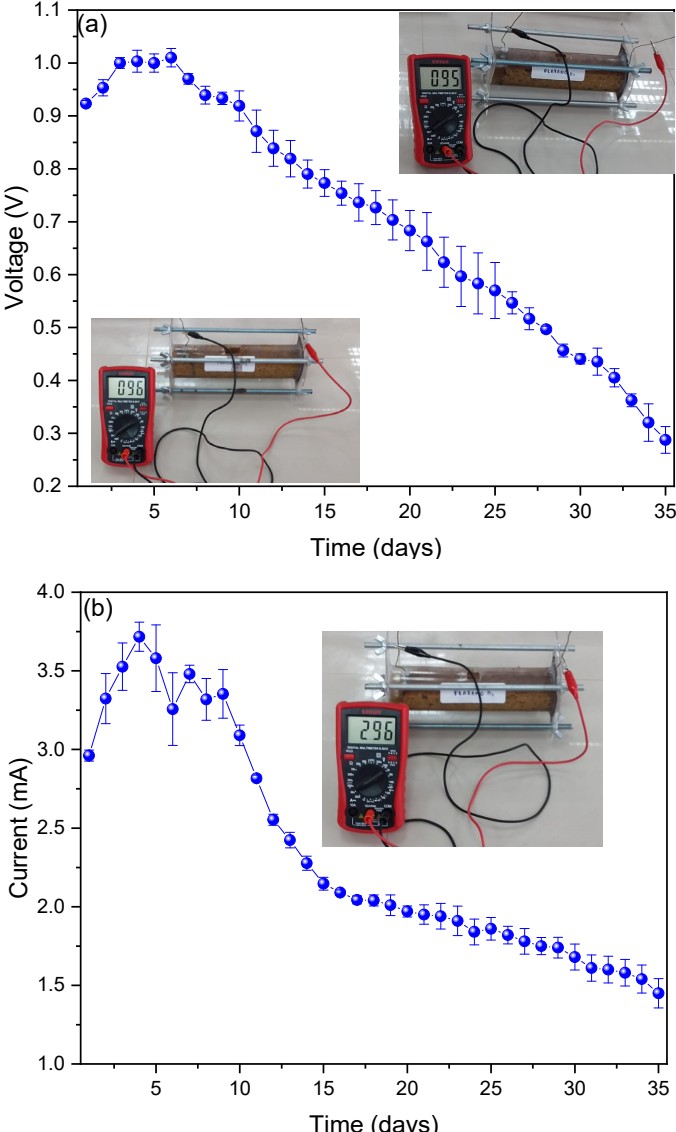

**Figure 2.** Values of (**a**) voltage and (**b**) electrical current monitored from microbial fuel cells.

The pH values are shown in Figure 3a. They increase from the first day, going from a moderately acidic to slightly acidic level, with an optimum pH of approximately

4.023 ± 0.064. The optimal pH value shown contradicts what is mentioned by Shukla and Kumar (2018), who mention that optimal pH should be between 6 to 8 normally, because it is optimal for the metabolic activity of microbes and any variation would produce a pH gradient reducing the values of electricity production [28]. However, it has been shown that the pH values are controlled by the biological activity associated with the microorganisms present in the cathodic and anodic biofilms, and not by the electrochemical performance of the microbial fuel cells [29]. Figure 3b shows the monitoring values of the electrical conductivity of the substrate, showing its maximum value on the seventh day (182.333 ± 3.51 µS/cm), which declined until the last day (43.667 ± 3.789 µS/cm).

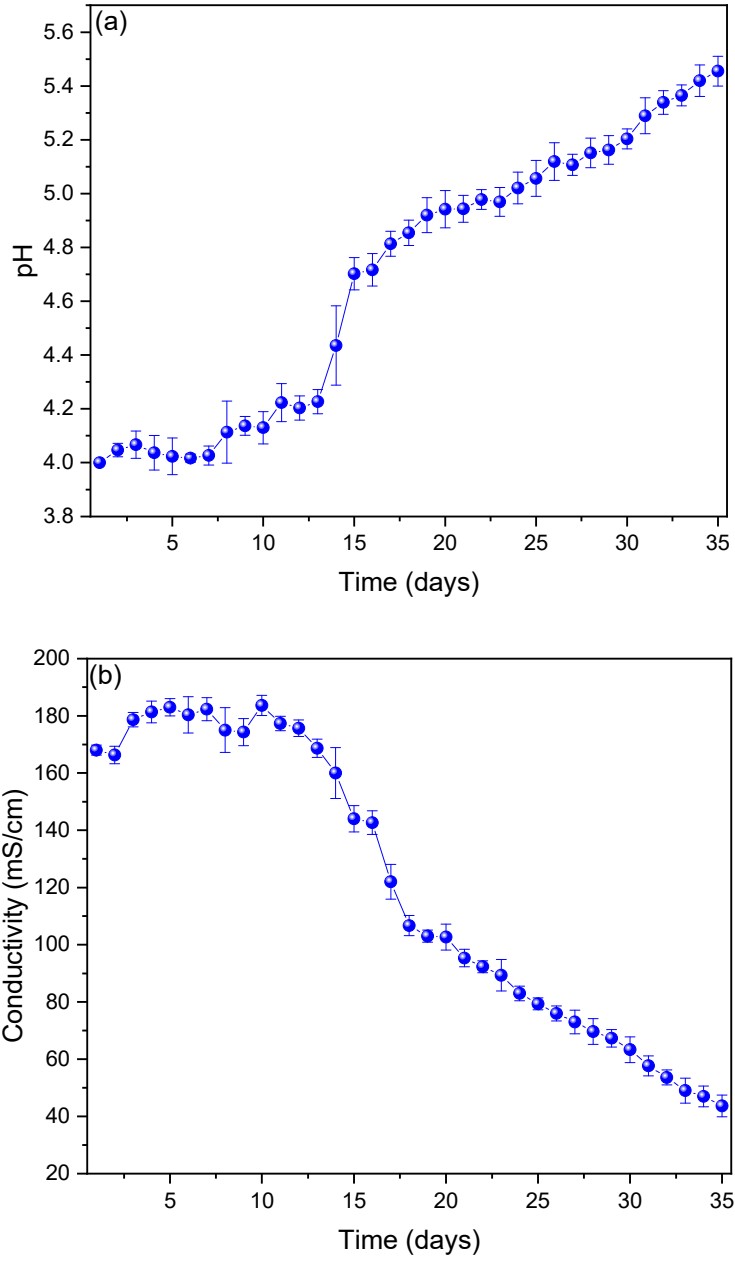

**Figure 3.** *Cont.*

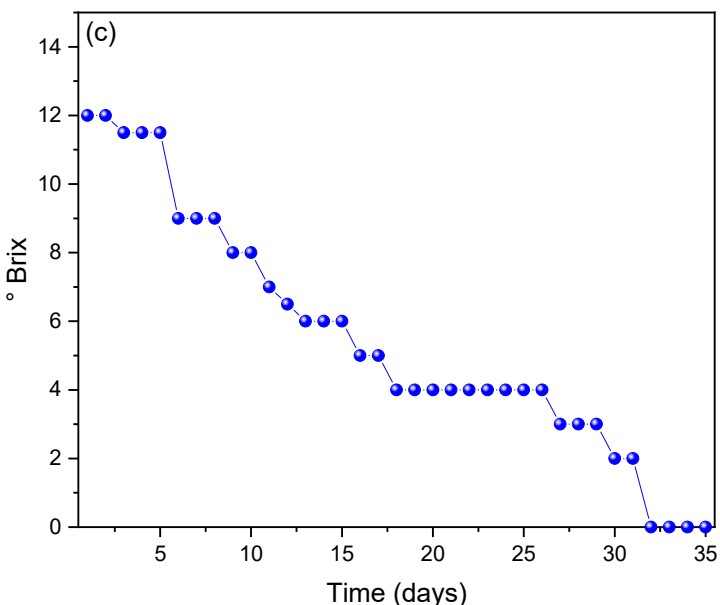

**Figure 3.** Monitoring of (**a**) pH, (**b**) conductivity, and (**c**) Brix values of microbial fuel cells.

The increase in conductivity values is mainly due to the proliferation of microorganisms in the first days of adaptation, while the decrease in values is due to the sedimentation of the substrate [30,31]. On the other hand, Figure 3c presents the Brix values generated by the cells, showing that from the first day (12 Brix) they decay slowly until the thirty-first day when they reach zero. Pazmiño et al. (2019) evaluated the feasibility of using banana stem residues as feedstock to produce biofuels such as ethanol and biogas from 0.591 g of juice/g of fresh stem composed of total soluble solids (Brix), such as glucose (7 g/L), sucrose (3 g/L), and fructose (8 g/L), obtaining, after five days of fermentation, 0.41 g of ethanol/g of sugars, because microorganisms consume sugars for the synthesis of new cellular components and metabolites [22,32]. Figure 4a shows the values of internal resistance ($R_{int.}$) of the microbial fuel cells, displaying that $R_{int.}$ remains almost constant throughout the monitoring with an average value of 152.43599 ± 5.654 Ω. This value shows a small decrease in the last minutes of the monitoring. In general, the high values of current and voltage generated are due to the low resistance shown by the system. The lower the resistance, the greater the passage of electrons, although in the final stage there is a small decrease in resistance that would lead to thinking that there is greater freedom for electrons to generate electric currents. This should be overshadowed, because, by that time, not as many electrons are generated as in the beginning due to the degradation and sedimentation of the substrate used [33–35]. Figure 4b shows the values of power density (PD) and voltage according to the current density (CD), showing a $PD_{MAX}$ of 5736.112 ± 12.62 mW/cm$^2$ at a CD of 6.501 A/cm$^2$ and a maximum voltage of 1006.95 mV. These values shown exceed those found by Kebaili et al. (2020), who used peeled fruits mixed with potassium chloride as a substrate, generating PD peaks of 0.12 mW/cm$^2$, mainly due to the use of fruit peel and low amounts of potassium chloride [36]. In the same way, a PD of 75 mW/m$^2$ was generated for cells with citrus peel substrate from which 350 mL of extract was obtained to be used as fuel [37].

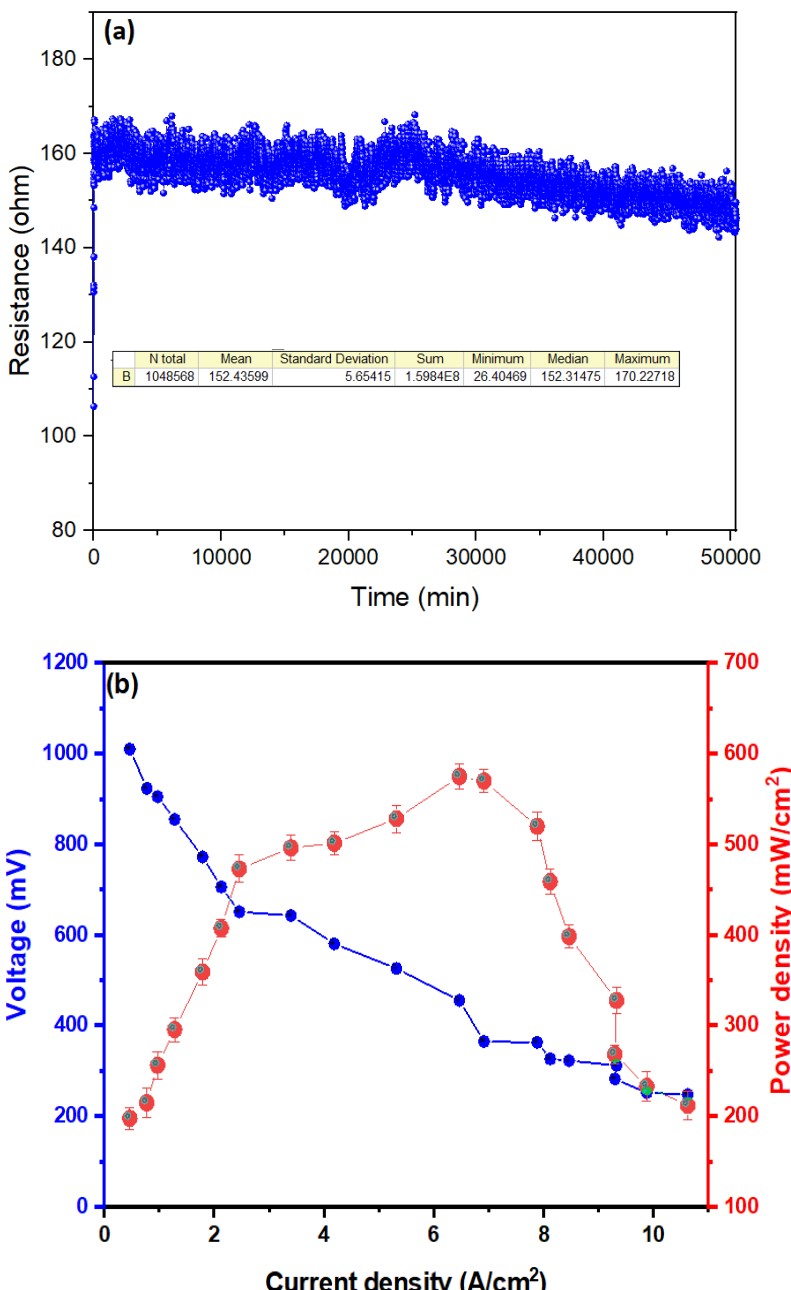

**Figure 4.** (**a**) Values of internal resistance and (**b**) power density according to current density.

Figure 5 shows the FTIR spectra of banana waste in its initial and final state. The most intense peak at 3367 cm$^{-1}$ belongs to the O-H bonds for phenolic compounds and 2938 cm$^{-1}$ belongs to the strong C-H bonds of the alkanes. Similarly, in the range 1639 cm$^{-1}$ the alkene compound was identified (C=C bond) and the peaks at 1415 and 1001 cm$^{-1}$ belong to the NO$_2$ and C-H bonds, respectively [38,39]. The peaks that are more noticeable in the spectrum are diminished in comparison to the final spectrum, which is due to the degradation of the substrate in the process of generating electrical energy during its operation, since the microorganisms use many of these compounds as food for their metabolism [40].

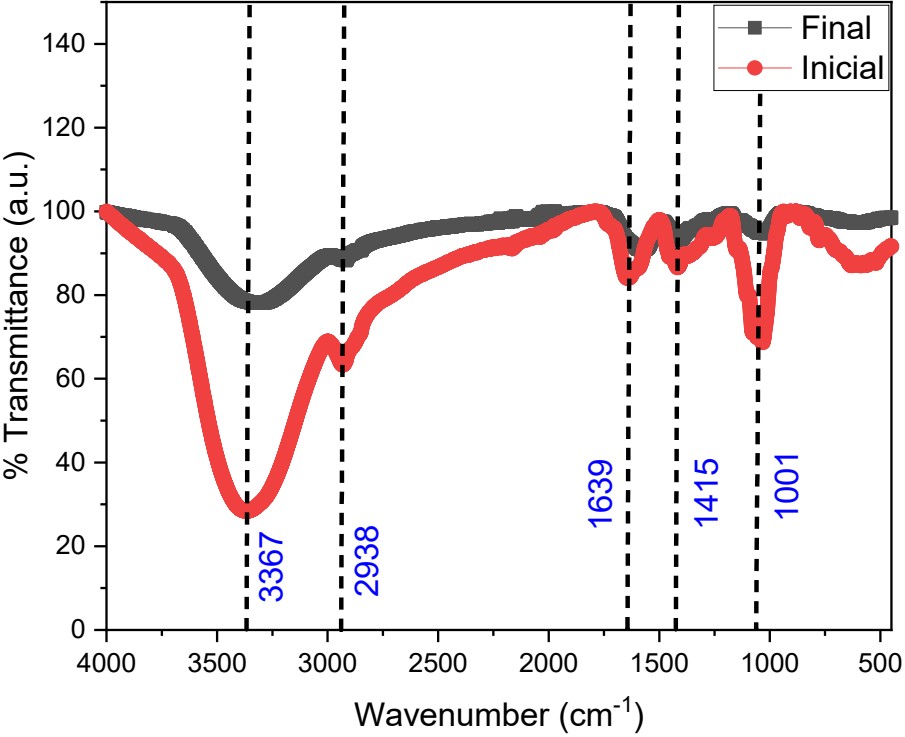

**Figure 5.** FTIR spectra of banana waste in its initial and final state.

Molecular identification of bacteria was achieved by sequencing a PCR product specific to the 16S bacterial ribosomal gene region [41]. They were analyzed by the BLAST program which obtained 100% of identity for the species *Pseudomonas aeruginosa*, 99.09% for the species *Paenalcaligenes suwonensis*, 99.39% for the species *Klebsiella oxytoca*, and 99.8% for the species *Raoultella terrigena* (Table 1).

**Table 1.** BLAST characterization of the rDNA sequence of bacteria isolated from the MFC anode plate with banana juice substrates.

| Sample Identification | BLAST Characterization | Length of Consensus Sequence (nt) | % Maximum Identity | Accession Number | Phylogeny |
|---|---|---|---|---|---|
| BANANA | *Pseudomonas aeruginosa* | 1442 | 100.00 | MT633047.1 | Cellular organisms; Bacterium; Proteobacterium; Gammaproteobacterium; Pseudomonadales; Pseudomonadaceae; Pseudomonas; *Pseudomonas aeruginosa* group |
| BANANA | *Paenalcaligenes suwonensis* | 1468 | 99.09 | NR_133804.1 | Cellular organisms; Bacterium; Proteobacterium; Betaproteobacteria; Burkholderiales; Alcaligenaceae; Paenalcaligenes |
| BANANA | *Klebsiella oxytoca* | 1468 | 99.39 | NR_118853.1 | Cellular organisms; Bacterium; Proteobacterium; Gammaproteobacterium; Enterobacter; Enterobacteriaceae; Klebsiella |
| BANANA | *Raoultella terrigena* | 1475 | 99.80 | LR131271.1 | Cellular organisms; Bacterium; Proteobacterium; Gammaproteobacterium; Enterobacter; Enterobacteriaceae; Raoultella |

A BLAST characterization of the rDNA sequence of bacteria isolated from the anode plate of microbial fuel cells with banana substrate was performed using ribosomal rDNA sequences, showing the levels of similarity between phylogenetically related species (see Figure 6). Among the species identified is *Pseudomonas aeruginosa*, a facultative aerobic bacterium [42] that can use carbon and nitrogen sources, obtaining energy from the oxidation of sugars. This species is persistent in the environment [43]. It is also worth mentioning that this species has electron mediators, such as phenazine-1-carboxylic acid and pyocyanin, that allow it to survive in anaerobic conditions [44]. A study by Ali et al. exposed the efficiency of using *Pseudomonas aeruginosa* species, which generated $136 \pm 87$ mW/m$^2$ using glucose, followed by fructose and sucrose. In this study, it was observed that the cell fed with glucose showed higher bacterial adhesion [45]. The species *Klebsiella oxytoca*, a Gram-negative bacterium and member of the Enterobacteriaceae family [46] present in the environment and humans [47], was also identified. It is worth mentioning that the genus Klebsiella is characterized by a prominent polysaccharide capsule [48]. The species *Paenalcaligenes suwonensis*, a Gram-negative, aerobic, catalase- and oxidase-positive bacterium, was also identified [49]. Another species identified was *Raoultella terrigena*, a Gram-negative bacterium of the genus Raoultella, isolated mainly from soil and water samples [50].

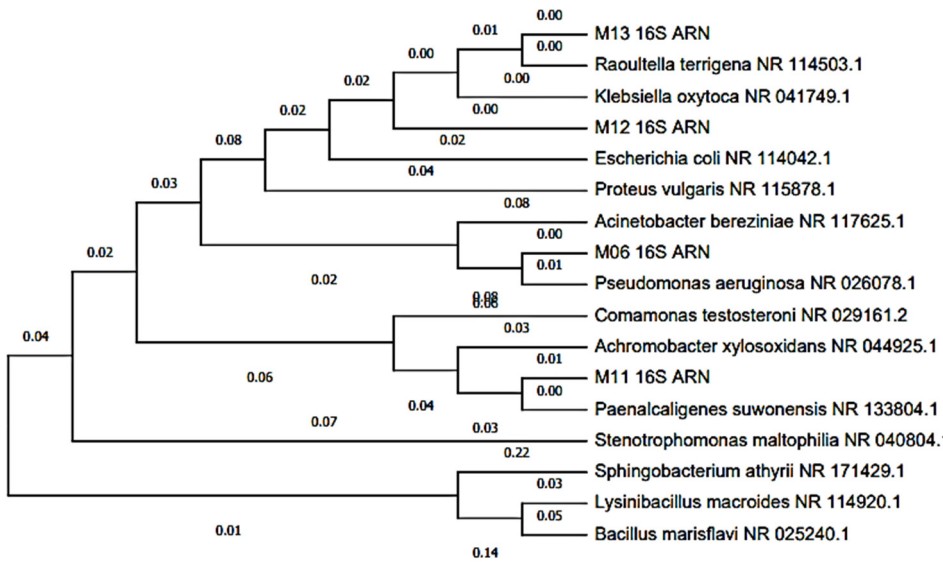

**Figure 6.** Dendrogram of bacterial clusters isolated from the MFC anode plate with banana substrate.

## 4. Conclusions

Bioelectricity was successfully generated by using banana waste as fuel through microbial fuel cells with zinc and copper electrodes at a laboratory scale. Peak voltages of $1.01 \pm 0.017$ V and $3.71667 \pm 0.05304$ mA were generated on the sixth and fourth day, respectively. The pH values increased from the first day, reaching an optimum pH of $4.023 \pm 0.064$ and electrical conductivity peaks of $182.333 \pm 3.51$ μS/cm on the seventh day. Likewise, the Brix values decreased from the first day until day 31, when they reached zero. The internal resistance of the system was $152.43599 \pm 5.654$ Ω, a very low value compared to other works, and its maximum power density was $5736.112 \pm 12.62$ mW/cm$^2$ for a current density of 6.501 A/cm$^2$ with a maximum voltage of 1006.95 mV. The FTIR spectra initially shows intense peaks of phenolic compounds, alkanes, and alkenes, among others. These peaks clearly decay with the final spectrum due to the process of bioelectricity generation. The species *Pseudomonas aeruginosa*, *Paenalcaligenes suwonensis*, *Klebsiella oxytoca*, and *Raoultella terrigena* were molecularly identified with an identity of 100, 99.09, 99.39, and 99.8%, respectively, which were found in the biofilm of the anode electrode. For future investigations, it is recommended to investigate with stable pH values (adding some chemical compound) and to cover the electrodes with non-toxic materials for microorganisms.

This research provides a new and eco-friendly way to use the waste of this fruit as fuel, which will be very beneficial for producers because they could use their waste to generate electricity for their own benefit.

**Author Contributions:** Conceptualization, S.R.-F.; methodology S.M.B.; software, R.N.-N.; validation, L.A.-S. and E.M.-T.; formal analysis, S.R.-F. and M.D.L.C.-N.; investigation S.R.-F.; data curation, M.D.L.C.-N.; writing—original draft preparation, D.D.-N.; writing—review and editing, S.R.-F.; project administration, S.R.-F. and R.N.-N. All authors have read and agreed to the published version of the manuscript.

**Funding:** This research was funded by Consejo Nacional de Ciencia, Tecnología e Innovación Tecnológica—CONCYTEC/PROCIENCIA according to Project agreement 370-2019.

**Institutional Review Board Statement:** Not applicable.

**Informed Consent Statement:** Not applicable.

**Data Availability Statement:** Not applicable.

**Acknowledgments:** The authors thank Juan Pastrana, for the help provided in the grammar of the English language.

**Conflicts of Interest:** The authors declare no conflict of interest.

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
