# Peer review of "Use of Banana Waste as a Source for Bioelectricity Generation"

_processes, doi:10.3390/pr10050942_

Round 1
Reviewer 1 Report
Comments and suggestions for authors are provided in attached file.

Author Response
Dear colleague, I hope you have a good day. The suggested changes have been made. Abstract 1. modified as reviewer suggest 2. modified as reviewer suggest Introduction 1. modified as reviewer suggest 2. modified as reviewer suggest 3. modified as reviewer suggest 4. modified as reviewer suggest best regards

Reviewer 2 Report
The authors propose an interesting topic that is in line with the Goals for Sustainable Development. This is futuristic research and should be taken into consideration for its contribution to improving the sustainability of the planet. Some minor revisions are attached to help researchers. The article deserves to be published.
Line 68 to 71, rewrite that sentence, it is not well understood.
Section 2 of Methodology and Materials should be expanded, perhaps some images of real experiments would help their understanding and improve the discussion. In addition, perhaps it is not necessary to abuse so many subsections.
In the results the size of the graphs must be increased, some are really difficult to interpret. Could the authors add an image of the experiment carried out?
Figure 4(a) is very cumbersome, the line thickness should be a little more careful.
The conclusions, although relevant, I think they can be expanded a little more indicating the implications of this research, its future lines of work and the possibilities of this technology in the future.
Author Response
Dear Colleague, the changes were made.
Only the Methodology and Materials part was not changed because the other reviewer suggested that it is fine. On the other hand, the thickness of the data in Fig. 4 (a) cannot be modified because the program gives it to me like this.
I hope you like this improved version.
best regards.
Good day
